# HDAC Inhibitors Can Enhance Radiosensitivity of Head and Neck Cancer Cells Through Suppressing DNA Repair

**DOI:** 10.3390/cancers16234108

**Published:** 2024-12-07

**Authors:** Jennifer Antrobus, Bethany Mackinnon, Emma Melia, Jonathan R. Hughes, Jason L. Parsons

**Affiliations:** 1Department of Molecular and Clinical Cancer Medicine, University of Liverpool, 6 West Derby Street, Liverpool L7 8TX, UK; jennifer.antrobus@manchester.ac.uk; 2Institute for Cancer and Genomic Sciences, University of Birmingham, Edgbaston, Birmingham B15 2TT, UK

**Keywords:** DNA damage, DNA repair, HDAC, histone, head and neck cancer, ionising radiation

## Abstract

Head and neck cancers are the seventh-most common cancers worldwide, with reported incidence of ~800,000 cases per year worldwide. Radiotherapy remains an important treatment for the disease, although cancer resistance remains a common problem. We performed a drug screen to identify those that can enhance the sensitivity of head and neck cancer models to radiotherapy. We identified that specific drugs (mocetinostat, CUDC-101, and pracinostat) can cause more effective cell killing of head and neck cancer cells grown in both 2D and 3D, and hold promise as more effective treatments for patients in the clinic.

## 1. Introduction

Head and neck squamous cell carcinoma (HNSCC) is the seventh-most common cancer worldwide, with an annual reported incidence of ~800,000 new cases and ~400,000 deaths every year [1]. The incidence of HNSCC is rising, and on average there has been a ~24% increase over the last decade [2]. Most HNSCCs arise in the epithelial linings of several regions in the head and neck. These include the oral cavity, nasal cavity, pharynx, larynx, salivary glands, and sinuses, and where the major risk factors for tumour development are tobacco and alcohol consumption. Additionally, infection with human papillomavirus (HPV) type 16/18 is a risk factor that accounts for ~60% of all tumours of the oropharynx [3,4]. HPV-positive oropharyngeal tumours are well established to display improved prognosis and survival rates than patients with HPV-negative tumours due to an enhanced response to radiotherapy and chemotherapy [5,6,7,8]. This differential response to radiotherapy based on HPV status has also been captured in cell lines derived from the respective tumours, where defective repair of ionising radiation-induced DNA double-strand breaks (DSBs) has been shown [9,10,11]. Therefore, the inherent radioresistance of HPV-negative HNSCC has become a target in an attempt to improve the treatment and outcomes of patients with this tumour subtype. Given that ionising radiation predominantly exerts its therapeutic effect through the damaging of DNA that drives tumour cell killing, DNA repair protein pathways are considered an important target for drugs or small-molecule inhibitors that can overcome HNSCC radioresistance. Indeed, evidence has demonstrated that inhibitors against enzymes such as poly(ADP-ribose) polymerase 1 (PARP-1), ataxia–telangiectasia mutated (ATM), ataxia telangiectasia and Rad3-related (ATR), and DNA-dependent protein kinase catalytic subunit (DNA-PKcs) can enhance the radiosensitivity of HNSCC cells, at least in vitro [12,13,14,15,16]. Despite this, other therapeutic targets and strategies that can overcome the radioresistance of HNSCC still need to be considered.

Genomic DNA in eukaryotic cells is wrapped around a histone octamer, which forms the basic level of chromatin [17]. The N-terminal tails of the histone proteins can be subject to a number of post-translational modifications that regulate and remodel the chromatin structure and therefore control DNA accessibility required for processes such as replication and transcription. Acetylation is one of the better-understood histone post-translational modifications and is dynamically controlled by histone acetyltransferases (HATs) and histone deacetylases (HDACs). Generally, acetylation by HATs weakens the histone–DNA interaction, causing chromatin decompaction and therefore activating gene expression. Conversely, HDACs promote deacetylation, thus restoring the strong histone–DNA interaction and leading to chromatin compaction and gene silencing. Histone acetylation has also been suggested to play a role in the regulation of DNA repair, particularly DSB repair [18,19]. HDAC inhibitors are therefore considered to have anti-cancer potential, but also to act as tumour radiosensitisers in response to different forms of radiotherapy [20,21]. Indeed, and as an example, valproic acid (VPA) has been shown to radiosensitise colorectal, oesophageal, and thyroid cancer cells to photon (X-ray) irradiation and hepatocellular carcinoma cells to protons [22,23,24,25]. Belinostat is thought to increase the sensitivity of rhabdomyosarcoma, cervical, and colorectal cancer cells to photon irradiation [26,27], while panobinostat can also radiosensitise bladder, prostate, and hepatocellular carcinoma cells [28,29]. Vorinostat has furthermore been shown to radiosensitise melanoma, lung, breast, and colorectal cancer cells to photons [30,31,32]. Whilst broadly, these studies demonstrate the potential for HDAC inhibition to effectively improve the sensitivity of cancer cells to radiotherapy, interestingly, this strategy has not been reportedly explored in HNSCC.

We hypothesized that current FDA-approved drugs exist that could be repurposed or reinvestigated as radiosensitisers of HNSCC. Therefore, we performed a targeted drug screen using a 3D spheroid model of HNSCC with the aim of identifying these novel HNSCC radiosensitisers. Through this screen and subsequent validation experiments, we discovered that the HDAC inhibitors mocetinostat and pracinostat and the combined HDAC–epidermal growth factor receptor (EGFR) inhibitor CUDC-101 improved the sensitivity of 3D and 2D models of HPV-negative HNSCC cells to X-ray irradiation. Furthermore, we elucidated that this radiosensitisation was a consequence of the inhibition of the repair of IR-induced DSBs.

## 2. Materials and Methods

### 2.1. Cell Lines and Culture Conditions

FaDu (hypopharynx) and A253 (salivary gland) cells came from ATCC (Teddington, UK), while UMSCC11b (larynx) cells were kindly provided by Professor T. Carey, University of Michigan, USA. All cell lines are HPV-negative and were routinely cultured in Dulbecco’s modified Eagle medium (DMEM), with the exception of FaDu cells, which were cultured in minimal essential medium supplemented with 10% fetal bovine serum, 1× penicillin–streptomycin, 2 mM L-glutamine, and 1× non-essential amino acids. All cells were maintained and incubated in 5% CO_2_ at 37 °C, and were authenticated in our laboratory by STR profiling.

### 2.2. Cell Irradiation

All irradiations were performed utilising a CellRad X-ray irradiator (Faxitron Bioptics, Tuscon, AZ, USA) at a standard dose rate of 3 Gy/min, 3 mA, and 100 kV. Doses from 1–4 Gy were given in a time-controlled manner.

### 2.3. Spheroid Drug Screen and Growth Assays

Spheroid culture and growth were conducted as previously described [12]. In brief, FaDu cells were seeded into 96-well ultralow-attachment round-bottomed plates at a density of 500 cells in 100 μL per well in advanced DMEM/F12 medium supplemented with 1% B27 and 0.5% N2 supplements, 2 mM L-glutamine, 1× penicillin–streptomycin, 5 µg/mL heparin, 20 ng/µL EGF, and 10 ng/µL FGF. After 24 h to allow spheroids to form, these were treated with DMSO (as a vehicle control) or compounds from an FDA-approved drug library (Selleck Chemicals GmbH, Planegg, Germany) for 24 h. For the drug screen involving 183 compounds (Appendix A), concentrations of both 0.03 and 1 µM drug were used, but only responses of single spheroids were analysed. For validation experiments, these consisted of three independent biological experiments that each contained triplicate spheroids. Following this, spheroids were treated with 1–4 Gy X-rays, 50 μL medium was removed from each well, and 100 μL fresh medium was added. The spheroids were therefore continually cultured in drug-containing medium for the duration of the experiment, albeit at a lower concentration than the initial dose. Spheroids were imaged between days 3 and 15 post-seeding using the EVOS M5000 imaging system (Life Technologies, Paisley, UK), and spheroid diameter (d) was measured using ImageJ (version 1.53). Diameters were converted into spheroid volume (V) using the formula V = 4/3 × π(d/2)^3^, and the fold increase in growth determined relative to the spheroids on day 3. Changes in spheroid growth as a function of X-ray dose was determined by analysing the fold increase in spheroid volume between days 3 and 15 post-seeding in the DMSO control versus the fold increases following the drug treatment.

### 2.4. Cell Viability and Clonogenic Survival Assays

For cell viability assays, 10,000 cells were treated with a serial dilution of mocetinostat, CUDC-101, or pracinostat (0.01–100 µM) in a 96-well plate 24 h after seeding using either DMSO as a vehicle-only control or hydrogen peroxide (10 mM) as a positive control. Cells were treated with these drugs for 72 h before viability was assessed. CellTiter Blue reagent (Promega, Fitchburg, WI, USA) was added and cells were incubated in 5% CO_2_ at 37 °C for 2 h before measuring absorbance at a wavelength of 570 nm (A_570_), with a reference wavelength of 600 nm (R_600_). A_570_–R_600_ was calculated, and the survival of drug-treated cells relative to the DMSO control (after subtraction of the positive control) was determined.

For clonogenic assays, cells were seeded in a defined number in triplicate in 24-well plates and left for 6 h to allow them to attach, and then fresh medium was added containing either DMSO (as a vehicle only control) or 1 µM drug. The plates were then incubated overnight before being treated with 1–4 Gy X-rays. Post-irradiation, the medium was changed and the plates were incubated for 7–14 days to allow colony (consisting of at least 50 cells) formation. Colonies were fixed and stained with 6% glutaraldehyde and 0.5% crystal violet for 30 min, washed, and then left to air-dry overnight. A GelCount colony counter (Oxford Optronics, Oxford, UK) was used to count colonies, and the surviving fraction was calculated by using the number of colonies per treatment level versus the number of colonies achieved in the untreated control acquired from three independent biological experiments. Statistical analysis of the differences across the treatment doses comparing the drug versus the DMSO control was performed using the CFAssay for R package [33], utilising the linear quadratic (LQ) model.

### 2.5. Neutral Comet Assay

The neutral comet assay was performed as previously described, analysing in-gel DNA repair activities [9]. In brief, cells following drug treatment for 24 h were trypsinised and diluted to 1 × 10^5^ cells/mL. Aliquots (250 μL) of the cell suspension were added to the wells of a 24-well plate on ice prior to X-ray irradiation (4 Gy). The cell suspension was mixed with 1% low-melting-point agarose, and this loaded onto slides that had been pre-coated and dried with 1% normal-melting-point agarose. The agarose was allowed to set for 2–3 min on ice, and slides then placed into a humidified incubator for up to 4 h to enable DNA repair. Slides were then submerged in cold lysis buffer (2.5 M NaCl, 100 mM EDTA, 10 mM Tris-HCl pH 10.5, 1% N-lauroylsarcosine, 1% DMSO, and 1% Triton X-100 (Thermo Fisher Scientific, San Francisco, CA, USA)) for at least 1 h at 4 °C to promote cell lysis. Slides were then washed with 1x TBE and placed in a darkened comet tank (Appleton Woods, Birmingham, UK) containing 1x TBE, where they were incubated for 30 min to enable DNA unwinding. Electrophoresis was performed at 25 V, ~20 mA for 25 min, after which the slides were washed three times with PBS and left to air-dry overnight. Following rehydration with water (pH 8) for 30 min, the electrophoresed DNA was stained with SYBR Gold (Life Technologies, Paisley, UK) diluted 1:20,000 in water (pH 8) for 30 min, and slides then allowed to air-dry. Cells were imaged using an Olympus BX61 microscope (Olympus, Shinjuku, Japan) at 10× magnification and analysed using the Komet 6.0 imaging analysis software (Andor Technology, Belfast, UK). Average percentage tail DNA values were determined from three independent biological experiments, and statistical analysis was performed using *t* tests.

### 2.6. Immunofluorescence Microscopy

Immunofluorescence staining to analyse γH2AX foci was performed as previously described [9]. In brief, cells were grown on 13 mm round glass coverslips and incubated with DMSO (as a vehicle-only control) or drug for 24 h. Cells were then treated with 4 Gy X-rays and incubated at 37 °C in 5% CO_2_ for up to 8 h post-irradiation to allow for DNA repair, prior to fixing with 10% formalin for 10 min at room temperature. To enable permeabilization, cells were treated with 0.2% Triton X-100 in PBS for 10 min. Following three washes with 0.1% Tween for 10 min, these were blocked with 2% BSA for 1 h at room temperature. Coverslips were then incubated overnight with γH2AX antibodies diluted 1:4000 (ab26350; Abcam, Cambridge, UK) at 4 °C. Following three washes with PBS, coverslips were incubated with goat anti-rabbit Alexa Fluor 488 secondary antibodies for 1 h at room temperature in the dark, washed for 10 min with PBS, and mounted on a microscope slide using Fluoroshield containing DAPI (Merck Life Sciences UK Ltd., Gillingham, UK). Cells were imaged using an Olympus BX61 microscope (Olympus, Shinjuku, Japan) at 40× magnification and captured using cellSens dimensions software (version 4.3). Average numbers of γH2AX foci per nuclei were determined from three independent biological experiments. Statistical analysis was performed using *t* tests.

### 2.7. Immunoblotting of Histone Acetylation

Cells were treated with DMSO (as a vehicle-only control) or drug for 24 h, and then left unirradiated or treated with 4 Gy X-rays and harvested 2 h post-irradiation. Acid extraction was used to purify histones. Cells were collected and then resuspended in hypotonic lysis buffer (10 mM Tris-HCl, pH 8.0, 1 mM KCl, 1.5 mM MgCl_2_) containing protease inhibitors (1 µg/mL each of pepstatin, chemostatin, leupeptin, and aprotinin, and 100 mM phenylmethylsulfonyl fluoride) prior to incubation for 30 min at 4 °C with shaking. Nuclei were pelleted by centrifugation (10,000× *g* for 10 min at 4 °C), resuspended in 0.4 M sulfuric acid, and incubated for 30 min at 4 °C with shaking. Following centrifugation (16,000× *g* for 10 min at 4 °C), the supernatant containing the histones was then mixed with 33% trichloroacetic acid and incubated for 30 min on ice. Following centrifugation (16,000× *g* for 10 min at 4 °C), histone proteins were washed twice with ice-cold acetone and then air-dried at room temperature. Histones were finally dissolved in water and analysed by immunoblotting using an Odyssey image analysis system (Li-cor Biosciences, Cambridge, UK). The following antibodies against unmodified or site-specific acetylated histones H3 and H4 were used at a dilution of 1:1000: H3 unmodified (3638); H3K27 (4353); H3K18 (9675); and H4K8 (2594) (all Cell Signaling Technology, Danvers, MA, USA).

## 3. Results

### 3.1. Targeted Drug Screen for HNSCC Radiosensitisers

From an FDA-approved drug library, 183 drugs were carefully selected for screening based on their targeting of key cellular processes and potential for anti-tumour activity and radiosensitising potential (Appendix A). These included pathways linked to cellular signalling and DNA synthesis, transcription, replication, and repair. A 3D spheroid model using HPV-negative FaDu cells (previously characterised [12]) was utilised for screening. Over the course of 15 days post-seeding and following drug and radiation treatment on days 2 and 3, respectively the fold change in spheroid volume as a function of growth was monitored to determine the radiosensitisation potential of the drugs (Figure 1A). Each drug was tested once at two concentrations (0.03 μM and 1 μM) in the absence and presence of a single 1 Gy dose of X-rays. To identify potential radiosensitisers, the increase in spheroid growth from days 3 to 15 post-seeding achieved with the drugs following 1 Gy X-rays post-irradiation was compared to that achieved following DMSO (as a vehicle control) post-irradiation which was normalised to 1.0 (Figure 1B,C; see also Appendix A). Interestingly, a significant proportion of the drugs tested caused an increase in the relative spheroid growth, which may have been mediated through enhanced radioresistance. Nevertheless, any drugs that reduced growth by at least 50% when compared to the DMSO control were taken into consideration, and then confirmed that effects were radiation specific by examining the impact of the drug versus the DMSO control alone. Through this system, we identified 17 candidate drugs potentially acting as radiosensitisers (Table 1), predominantly involved in cell growth, survival, and signalling, as well as chromatin organization, cell cycle regulation, and DNA/RNA synthesis. Interestingly, this identified epidermal growth factor receptor (EGFR) as a prominent target, although recently failed trials using cetuximab have cast doubt over EGFR targeting as a therapeutic strategy in HNSCC [34,35]. However, several HDAC inhibitors were also identified as potential radiosensitisers, which were specifically taken forward from the screen.

### 3.2. Evaluation of Candidate HNSCC Radiosensitisers

The screen (performed using an individual replicate) identified that the class I-selective HDAC inhibitor mocetinostat, the broad-spectrum HDAC inhibitor pracinostat, and the broad-spectrum HDAC–EGFR inhibitor CUDC-101 appeared to radiosensitise and suppress growth of 3D FaDu spheroids. To further validate these data, we initially carried out additional spheroid experiments using a set drug dose (0.2 µM) aiming to ensure sufficient HDAC inhibition without stimulating cell toxicity. This revealed that the combination of mocetinostat with a 2 Gy dose of X-rays had minimal impact on the growth of FaDu spheroids over a 15 day period post-seeding compared to the radiation alone, whereas with a 4 Gy dose, there was a ~22% decrease in spheroid volume (Figure 2A,D). In comparison, the combination of either CUDC-101 or pracinostat with X-rays suppressed spheroid growth by 20–27% in comparison to the 2 and 4 Gy radiation doses alone (Figure 2B,C,E,F). However, it was noticeable that the inhibitors alone also led to an impact on spheroid growth compared to the DMSO control. Therefore, in order to determine whether the inhibitors were acting synergistically with radiation, we analysed the change in spheroid growth from days 3 to 15 relative to radiation dose. This revealed that only CUDC-101 and pracinostat, but not mocetinostat, demonstrated significantly increased radiosensitivity compared to spheroids treated with DMSO alone (Figure 2G–I), and so these drugs were acting synergistically with X-rays.

To extend from this evidence, we analysed the toxicity of increasing doses of the three drugs on FaDu cells grown as monolayers. Cell viability assays demonstrated that the cells were resistant up to 10 µM doses of mocetinostat, CUDC-101, and pracinostat, whereas a dose of 100 µM was cytotoxic (Figure 3A–C). Alternatively, using clonogenic assays, this revealed that FaDu, A253, and UMSCC11b (all HPV-negative HNSCC cells) were able to tolerate 1 µM doses of the drugs, although notable inhibition of clonogenic cell survival was observed with CUDC-101 in FaDu cells and mocetinostat in UMSCC11b cells (Figure 3D–F). We also confirmed that all four HDAC inhibitors (at 1 µM doses) were effective in suppression of histone deacetylase activity, evident through enhanced acetylation of site-specific residues on histones H3 and H4 in FaDu cells, independently of X-ray irradiation (Figure 3G).

### 3.3. Confirmation of HDAC Inhibitors as HNSCC Radiosensitisers

To support and extend from evidence using 3D spheroid models, we analysed clonogenic survival (the gold-standard assay in radiobiology) of FaDu, along with two other HPV-negative HNSCC cell lines, A253 and UMSCC11b, in the presence of the HDAC inhibitors following increasing doses of X-rays. We observed that all three drugs were able to significantly enhance the radiosensitivity of the cells compared to DMSO treatment alone (Figure 4A–F). Statistical analysis revealed cellular radiosensitisation by mocetinostat, CUDC-101, and pracinostat was statistically significant in FaDu, A253, and UMSCC11b cells. We also calculated the dose enhancement ratios (DERs) compared to DMSO treatment alone, which revealed enhanced radiosensitisation by mocetinostat (DER = 1.37–1.82), CUDC-101 (DER = 1.18–1.91), and pracinostat (DER = 1.35–1.73) across the cell lines. Collectively, this provides evidence that the HDAC inhibitors mocetinostat, CUDC-101, and pracinostat can sensitise HPV-negative HNSCC cells to X-ray irradiation.

### 3.4. Impact of HDAC Inhibitors on DNA Damage Repair

Histone acetylation coordinated by HDACs can control DNA-dependent processes, including DNA repair. Therefore, we analysed the rates of DSB repair in the presence of mocetinostat, CUDC-101, or pracinostat post-irradiation using the neutral comet assay as a direct detection of DSBs and through γH2AX foci as a surrogate DSB marker using immunofluorescence staining. In FaDu cells following DMSO treatment, the majority of radiation-induced DSBs were found to have been repaired by 4 h post-irradiation, as demonstrated by the comet assay through percentage tail DNA, where the levels reverted to those similar to the unirradiated control (Figure 5A,E). However, treatment with mocetinostat, CUDC-101, or pracinostat prior to irradiation led to statistically significant persistence in the levels of DSBs at 2 and 4 h post-irradiation compared to the DMSO control-treated cells. Interestingly, the inhibitors also caused a significant increase in the DSB levels in control samples in the absence of X-ray irradiation. Similar data were acquired in A253 cells, evidenced through persistence of both DSB levels at 2–4 h post-irradiation in the presence of mocetinostat, CUDC-101, or pracinostat compared to DMSO-treated cells (Figure 5B,F). These data are supported by analysis of the surrogate DSB marker γH2AX, where there was evidence of the persistence of γH2AX foci at 4–8 h post-irradiation in the presence of the HDAC inhibitors compared to the DMSO-treated cells in both FaDu (Figure 5C,G) and A253 cells (Figure 5D,H). Collectively, this provides evidence that that the class I HDAC inhibitor mocetinostat, the pan-HDAC inhibitor pracinostat, and the broad-spectrum HDAC–EGFR inhibitor CUDC-101 induced radiosensitisation in HNSCC cells through their impact in suppressing the efficiency of DSB repair.

## 4. Discussion

Radiotherapy is commonly utilised as a treatment for HNSCC; however, tumour radioresistance remains a barrier to effective treatment, particularly in HPV-negative subtypes, which contributes to poor patient outcomes. This could be related to inherent resistance of the tumour cells caused by alterations in cellular processes, such as efficiencies of DNA repair, but also other intrinsic factors such as tumour hypoxia [36]. Therefore, strategies involving combinatorial treatments are necessary in order to improve the response and outcomes of HNSCC patients to radiotherapy. Due to the therapeutic effect of radiotherapy largely being achieved through causing DNA damage, there has naturally been a focus on targeting the DNA damage response in biological experiments using cell- and animal-based models [13,14,16]. However, alternative targets and strategies are considered necessary to further improve radiotherapy efficacy in HNSCC.

With this in mind, we performed a 3D spheroid-based screen of 183 FDA-approved drugs to identify those that could potentially be repurposed as novel HNSCC radiosensitisers. Spheroid models are highly beneficial for screening in comparison to 2D monolayer culture, as they are a more accurate representation of the 3D biological structure of tumours and how these respond to treatment [37]. Through our screen, we identified a number of HDAC inhibitors as potential radiosensitisers of FaDu spheroids. Through validation experiments utilising spheroid growth assays in FaDu cells and subsequent clonogenic survival assays in multiple HPV-negative HNSCC cell lines (FaDu, A253, and UMSCC11b), we confirmed that the broad-spectrum HDAC inhibitor pracinostat and the multi-targeted EGFR and HDAC inhibitor CUDC-101 can potentiate the effects of X-rays in HNSCC cell killing. Interestingly, we observed that the class I-specific HDAC inhibitor mocetinostat was additive in radiosensitising FaDu cells grown in 3D, whereas this was acting synergistically through clonogenic survival assays in all three HNSCC cell lines. All inhibitors were nevertheless proven to enhance the levels of histone acetylation at least on 2D monolayers, demonstrating their ability to suppress HDAC activity. Further experiments are required to fully understand why mocetinostat was not acting synergistically with X-rays in inhibiting growth of spheroids post-irradiation compared to experiments performed in 2D. It is difficult, though, to draw more definitive conclusions based on data using one spheroid model grown in 3D and where the drug is present throughout the spheroid growth period compared to clonogenic assays where cells are seeded and treated individually and the drug is replaced immediately post-irradiation. Therefore, the next steps would be to employ additional spheroid models, but also confirming HDAC inhibition across the cells within the 3D structure to confirm drug penetrance and activity. We are also keen to further investigate other HDAC inhibitors (such as resminostat, which appeared positive from our screen at the low drug dose), which may help to define the specific nature and target of the radiosensitisation. Nevertheless, we demonstrated that mocetinostat, CUDC-101, and pracinostat all lead to delays in radiation-induced DSB repair through inhibition of histone deacetylation, providing a mechanism of action whereby the drugs are likely to prevent sufficient access or coordinated action of the DNA repair proteins to the DSB damage in chromatin.

Interestingly, and at least through neutral comet assay analysis to directly detect DSBs, we observed that the HDAC inhibitors in the absence of X-ray irradiation were able to increase the levels of DSBs, suggesting that the drugs were able to also suppress the repair of endogenously induced DSBs. Previous work has provided links between changes in histone acetylation and the cellular DNA damage response either through the control of the expression of key DNA repair proteins or via regulating chromatin structure [20,21]. For example, HDAC1 and HDAC2 have been shown to swiftly localise to sites of IR-induced DNA damage in U2OS cells, where they downregulate histone H3 lysine 56 acetylation, and depletion of HDAC1/2 has been observed to cause hypersensitivity to IR through causing defects in DSB repair, particularly non-homologous end joining [38]. In mouse embryonic fibroblasts, HDAC3 inactivation increases sensitivity to the DNA-damaging agents doxorubicin and cisplatin, but this appears to be a result of changes in chromatin structure, as HDAC3 is not directly recruited to sites of DNA damage and does not affect the localisation of DDR proteins, including RAD51, BRCA1, and MRE11 [39,40]. HDAC4, on the other hand, has been shown to localise to nuclear foci post-irradiation in HeLa cells, where it co-localises with 53BP1 at DSB sites, and siRNA-mediated knockdown of HDAC4 enhances sensitivity of the cells to IR [41]. Finally, there is evidence to suggest that HDAC9 and HDAC10 are required for DSB repair by homologous recombination, but the mechanism involved has not yet been identified [42]. Nevertheless, our data acquired through the use of the class I-specific HDAC inhibitor mocetinostat as well as the broad-spectrum HDAC inhibitors CUDC-101 and pracinostat negatively impacting DSB repair efficiency in HPV-negative HNSCC cells would correlate with this previous evidence.

More specific to our research, a report has shown that mocetinostat can enhance the radiosensitivity of bladder cancer cells, which appeared to be mediated through reducing the protein levels of MRE11, RAD51, and NBS1 involved in DSB repair [29]. The radiosensitisation of pancreatic cancer cells with CUDC-101 has also been observed, although directs links with DNA damage repair were not demonstrated [43]. As previously mentioned, CUDC-101 is both a multi-targeted EGFR and HDAC inhibitor; however, given that EGFR targeting in HNSCC clinically has failed [34,35], combined with our observations that mocetinostat and pracinostat (and possibly other HDAC inhibitors included in the spheroid screen) are equally effective, if not more, in radiosensitising HNSCC cells in vitro, then we would advocate that HDAC inhibition is the key driver and should be a strategy for further investigation. Therefore, for the next steps, it will be important to further investigate the synergistic action of mocetinostat, pracinostat, and CUDC-101 with X-rays using a number of HNSCC spheroid models and eventually using more complex models such as patient-derived organoids. It would also be interesting to develop a more in-depth understanding of the precise histone acetylation sites targeted by the individual HDAC inhibitors that are crucial for radiosensitisation and DSB repair inhibition, as well as the specific mechanism linking these events (such as altered chromatin accessibility or DNA repair protein expression) in order to appreciate the full therapeutic potential of HDAC inhibition for the radiosensitisation of HNSCC. Nevertheless, we would advocate that our data represent an alternative strategy for HNSCC treatment that requires more detailed preclinical investigation.

## 5. Conclusions

Our study has identified the potential for the HDAC inhibitors mocetinostat, CUDC-101, and pracinostat in enhancing the efficacy of radiotherapy in the treatment of HNSCC models through the suppression of the repair of DNA DSBs. HDAC inhibitors with radiotherapy should therefore be considered as a combinatorial therapy for HNSCC patients and to overcome the treatment resistance that is commonly observed, particularly in HPV-negative disease. Further studies are required in more advanced preclinical models (patient-derived organoids and mouse models).

## Figures and Tables

**Figure 1 cancers-16-04108-f001:**
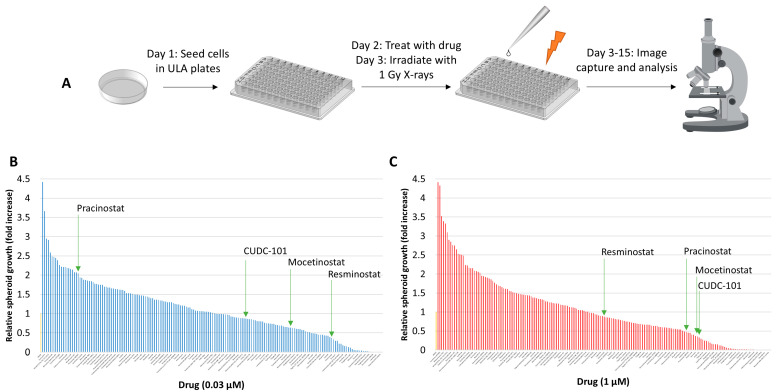
Screening of a selected number of FDA-approved drugs in the radiosensitisation of HNSCC cells. (**A**) Schematic of the drug-radiation screen utilising FaDu cells grown as 3D spheroids and then analysing growth over a period of 3–15 days post-seeding. Analysis of FaDu spheroid growth between 3 and 15 days post-seeding after a single 1 Gy dose of X-rays in the presence of 183 drugs at either (**B**) 0.03 µM or (**C**) 1 µM dose. Growth was analysed from a single replicate and normalised against the DMSO-treated control (first lane, yellow bar), which was set to 1.0. Indicated in green arrows are the HDAC inhibitors identified as potential radiosensitisers.

**Figure 2 cancers-16-04108-f002:**
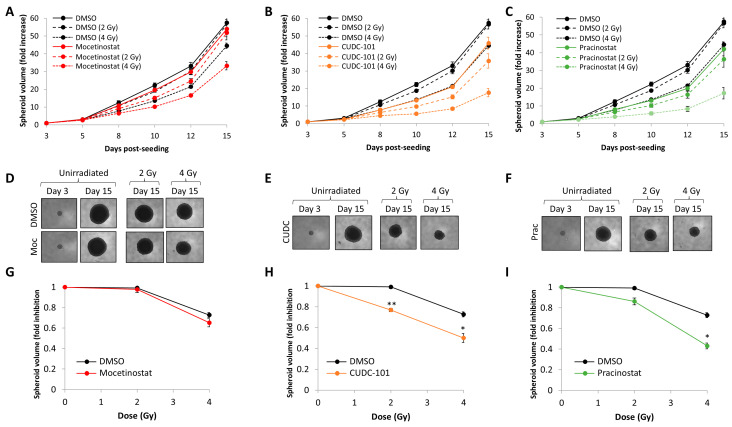
Validation of mocetinostat, CUDC-101, and pracinostat in suppressing HNSCC spheroid growth in combination with X-ray irradiation. Growth of FaDu spheroids in the presence of (**A**) mocetinostat, (**B**) CUDC-101, and (**C**) pracinostat (all 0.2 µM) alone, and following a 2 Gy or 4 Gy dose of X-rays in comparison to the DMSO-treated control. Shown is the mean fold increase in spheroid volume ± SE across three independent experiments normalised against the volume observed on day 3 post-seeding, which was set to 1.0. (**D**–**F**) Images of the spheroids with the different treatments on days 3 and 15 post-seeding. (**G**–**I**) Fold inhibition in spheroid volume ± S.E as a function of radiation dose determined between days 3 and 15 post-seeding across three independent experiments normalised to the respective unirradiated control, which was set to 1.0. * *p* < 0.05, ** *p* < 0.002 on two-sample *t*-tests.

**Figure 3 cancers-16-04108-f003:**
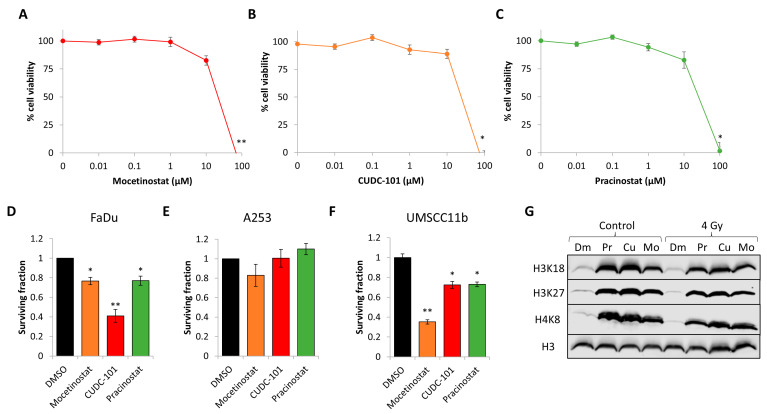
Effect of mocetinostat, CUDC-101, and pracinostat on HNSCC cell viability, clonogenic survival, and histone acetylation. (**A**–**C**) FaDu cells were treated with increasing doses of (**A**) mocetinostat, (**B**) CUDC-101, or (**C**) pracinostat (0.01–100 µM), and cell viability was measured via CellTiter Blue from three biologically independent experiments. Shown is the mean cell viability ± SE. * *p* < 0.05, ** *p* < 0.01 on two-sample *t*-tests compared to the untreated controls. (**D**) FaDu, (**E**) A253, and (**F**) UMSCC11b cells were treated with mocetinostat, CUDC-101, or pracinostat (all 1 µM) in comparison to DMSO as control and clonogenic survival of cells, analysed from three independent experiments. * *p* < 0.01, ** *p* < 0.001 on one-sample *t*-tests compared to the DMSO-treated controls. (**G**) FaDu cells were treated with DMSO (Dm), 1 µM mocetinostat (Mo), CUDC-101 (Cu), or pracinostat (Pr), and either unirradiated (control) or irradiated with 4 Gy X-rays and cells harvested at 2 h post-irradiation. Histones were purified by acid extraction and analysed by immunoblotting using antibodies targeted against site-specific acetylation sites on histone H3 or H4.

**Figure 4 cancers-16-04108-f004:**
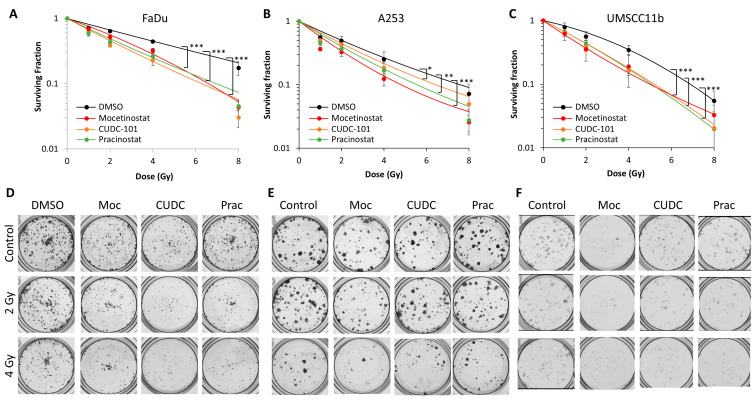
Analysis of the impact of HDAC inhibitors on the clonogenic survival of HNSCC cells post-irradiation. (**A**,**D**) FaDu, (**B**,**E**) A253, and (**C**,**F**) UMSCC11b treated with mocetinostat, CUDC-101, or pracinostat (all 1 µM) in comparison to DMSO as a control and clonogenic survival of cells with increasing doses of X-ray irradiation was analysed with three independent experiments. Shown is the mean surviving fraction ± SE. * *p* < 0.05, ** *p* < 0.0005, *** *p* < 0.0001, analysed with the LQ model.

**Figure 5 cancers-16-04108-f005:**
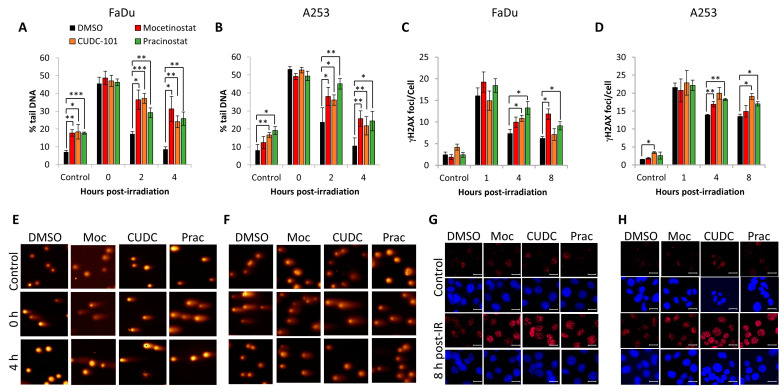
Mocetinostat, CUDC-101, and pracinostat cause delays in DSB repair in HNSCC cells. (**A**,**C**,**E**,**G**) FaDu and (**B**,**D**,**F**,**H**) A253 cells were treated with 1 µM CUDC-101, pracinostat, or DMSO as a control, and either unirradiated (control) or irradiated with 4 Gy X-rays and cells harvested at the various time-points post-irradiation. (**A**,**B**) Levels of DSBs were analysed directly using the neutral comet assay, with mean percentage tail DNA ± SE determined from three independent experiments. (**C**,**D**) Numbers of γH2AX foci were determined using immunofluorescence microscopy, with mean γH2AX foci/cell ± SE determined from three independent experiments. * *p* < 0.05, ** *p* < 0.01, *** *p* < 0.001 on one-sample *t*-tests. (**E**,**F**) Stained DNA in cells either unirradiated or 4 h post-irradiation following inhibitor treatment through gel electrophoresis. (**G**,**H**) γH2AX foci in cells either unirradiated or 8 h post-irradiation following inhibitor treatment. Scale bar is 20 µm.

**Table 1 cancers-16-04108-t001:** Potential HNSCC radiosensitisers identified in the spheroid-drug screen.

Drug	Target	Category
Stearic Acid	NF-kB	Cell Growth and Survival
Orantinib (TSU-68, SU6668)	PDGFR	
Apremilast (CC-10004)	TNF-alpha, PDE	
Mocetinostat (MGCD0103)	HDAC	Chromatin Organisation
Pracinostat (SB939)	HDAC	
Resminostat	HDAC	
Icotinib	EGFR	Cell Signalling
Rociletinib (CO-1686, AVL-301)	EGFR	
Lidocaine Hydrochloride	EGFR	
CUDC-101	EGFR, HDAC, HER2	
Afatinib (BIBW2992)	EGFR, HER2	
Genistein	EGFR, Topoisomerase	Cell Signalling/DNA Replication
Dexrazoxane HCl (ICRF-187, ADR-529)	Topoisomerase	DNA Replication
Novobiocin Sodium	Topoisomerase	
Enoxacin	Topoisomerase	
Fidaxomicin	DNA/RNA Synthesis	DNA/RNA Synthesis
Mupirocin	DNA/RNA Synthesis	

## Data Availability

The data presented in this study are available on request from the corresponding author.

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
