# Peer review of "HDAC Inhibitors Can Enhance Radiosensitivity of Head and Neck Cancer Cells Through Suppressing DNA Repair"

_cancers, 2024, doi:10.3390/cancers16234108_

Round 1
Reviewer 1 Report
Comments and Suggestions for Authors
In the present manuscript, Antrobus et al. evaluate HDAC as radiosensitizers in HPV-negative HNSCC. The authors perform drug screens at two doses and identify three HDAC inhibitors, namely, mecotinostat, pracinostat and the combined EGFR and HDAC inhibitor CUDC-101 as candidate radiosensitixers. Subsequently, the authors seek to further confirm radiosesntization by these drugs through spheroid formation and clonogenic assays. Lastly, the authors evaluate the effect of these drug candidates on enhancing DNA damage upon treatment with IR. Overall, there are a few issues related to the selection of drug candidates and their evaluation as well as figure annotation. These are outlined in the comments below:
1) Figure 1: In the plots, the names of the drugs in the X-axis are not legible. It is difficult to delineate the comparisons between relative growth of spheroids. One can assume that the initial bar showing relative growth = 1 relates to the DMSO control. However, it is impossible to tell whether this is an initial reading to which all spheroid growths are normalized or if the second bar represents the true growth under non-treated conditions. One may assume the latter is true given this is the only way a growth reduction of 50%, which the authors use to nominate drug targets makes sense. This is important to be rectified and represents a significant roadblock to proceeding in understanding the manuscript.
2) The authors pick pracinostat, which shows a relatively modest growth defects at lower doses. This is as opposed to Resminostat, which shows potency at low doses (albeit not as high at the higher dose condition). Thus, Resminostat could represent a desirable candidate owing to its potential avoidance of dose induced toxicity. The authors should clarify their reasons to exclude it from further analyses.
3) Figure 2G: Mocetinostat shows no growth effects at 2Gy radiation. This is at odds with the observation in Fig 2A where at 2Gy, Mocetinostat shows a notable growth inhibition. These differences in observations should be reconciled, especially if the data were derived from the same experiment.
4) Line 252: The wrong figure is referenced here.
5) Table 1: The percent growth inhibition for all of the described candidates should be provided in order to further clarify the method by which they were selected.
6) Figure 5: given the fact that Mocetinostat causes a greater impact in DNA damage enhancement in comet assays in FaDu and A253 cells at later timepoints and in yH2AX assays in FaDu cells it becomes a little problematic to draw the conclusion that DNA repair inhibition is a major mechanism of radiosensitization, especially given the only modest effect of Mocetinostat in prior experiments. While the authors do discuss difficulties in assessing the precise mechanisms of sensitization in the discussion, this particular observation should be expanded on.
Minor points:
1) The labels for Figure 2C are missing.
Reviewer 2 Report
Comments and Suggestions for Authors
In general, the manuscript entitled ‘HDAC inhibitors can enhance radiosensitivity of head and 2 neck cancer cells through suppressing DNA repair’ is clearly written. The main goal is thoroughly explained. Based on recent research, the authors present a gap in the literature that they try to fill with their analysis. This manuscript is clear and wort-reading. The findings are important and will be interesting to the readers of the journal. Specific comments and questions are provided below.
1. How was the drug treatment time optimised? I wonder why some analysis was performed after 2 hours, whereas others after 4h or 8 hours after irradiation.
2. In Sections 2.6 and 2.7 the catalogue numbers, and the dilutions of the antibodies (H2AX, unmodified, or site-specific acetylated histones H3 and H4 and secondary) should be added.
3. The authors conducted a study of 183 drugs in concentrations 0.03 and 1µM in the spheroid. Clarify why in further analysis the 0.2 µM drug dose was used.
4. In line 252 there should be probably Figure 2B-C and E-F (you’ve got Figure 3B-C and E-F).
5. Why did the authors use a 2 and 4 Gy dose of X-rays in the spheroid model if in the first analysis you tested 1Gy?
6. There is a mistake in the legends of Figures 2A and 2B. The X-ray dose on the histogram is 1 and 2 Gy, whereas in the description it is 2 and 4Gy.
7. Please add a legend to Figures 2C, 4B, and 4C.
8. Please explain why drug concentration 1µM was used for further experiments. I don’t see statistical significances between 0.01, 0.1, 1 or 10 µM concentration of the drugs tested.
9. Based on many studies suggesting that cells with different genetic profiles respond differently to the same drugs, the effect of mocetinostat, CUDC-101, and pracinostat on cell viability of A253 and UMSCC11b cells should also be tested.
10. Why is the H3 histone a control for H4K8 level? In my opinion, the H4K8 level should be compared to the H4 level (which was not analysed). Moreover, determining the level of the analysed proteins in one repetition is unreliable. The authors must consider whether they want to repeat this study or remove this result from the manuscript. If the result remains in this article, the histone modification level must be calculated and added to the graphic or presented in the graph.
11. I don't see any statistical significance in Figure 4. Perhaps the images are too small or the authors didn't put labels on the histograms. But the authors mentioned in line 304 that: Statistical analysis showed that cellular radiosensitization by mocetinostat, CUDC-101, and pracinostat was highly statistically significant (p<0.00001) in FaDu, A253, and UMSCC11b cells. This is misleading.
12. The results shown in Figure 3D-F should also be analyzed using a statistical test.
13. Student's t-test is used for pairwise comparisons only. Why was it used in the study if the authors have 3 or more groups? Using the Student's t-test in this case is inappropriate and generates false conclusions.
Reviewer 3 Report
Comments and Suggestions for Authors
The article is interesting, but some points can be improved. Thus, the study needs a minor revision.
- Simple Summary and Introduction: Is HNSCC the sixth or seventh most common cancer?
- Abstract: The authors must include the cell lines used and the analyzes performed.
- Introduction: Authors should include the study hypothesis and not the results in the last paragraph of the Introduction.
- Materials and Methods: The authors have carried out many experiments and in order to facilitate the reader's understanding, please consider including a flowchart with the steps of the methodology
- Materials and Methods: item 2.2: What is the reference for the irradiation dose?
- Results: In all figures, authors must indicate with symbols where there were significant differences.
- Discussion: authors must indicate the limitations of the study and the possible limitations/problems of using HDAC inhibitors in cancer treatment.
Round 2
Reviewer 2 Report
Comments and Suggestions for Authors
Thank you very much to the authors for comprehensive explanations and supplementing some missing information. In response to this, the quality of the manuscript increased significantly. Now, the methodology is well described and has so many details that the experiments can be easily replicated. For the future, please remember about statistical analyses and adding significance markers in the charts. This shows the validity of your research. Moreover, try to justify the choice of cell treatment conditions in the text. This will allow readers to more easily follow the researcher's thinking.
Author Response
We thank the Reviewer for their kind comments and suggestions.